# Assessment Tools Measuring Fundamental Movement Skills of Primary School Children: A Narrative Review in Methodological Perspective

**DOI:** 10.3390/sports11090178

**Published:** 2023-09-07

**Authors:** Ágnes Virág Nagy, Márta Wilhelm, Mihály Domokos, Ferenc Győri, Tamás Berki

**Affiliations:** 1Doctoral School of Biology and Sport Biology, University of Pécs, 7624 Pécs, Hungary; nagy.agnes.virag@szte.hu (Á.V.N.); mwilhelm@gamma.ttk.pte.hu (M.W.); 2Physical Education and Sports Sciences, ‘Juhász Gyula’ Faculty of Education, University of Szeged, 6725 Szeged, Hungary; domokos.mihaly@szte.hu; 3Institute of Sport Sciences and Physical Education, Faculty of Sciences, University of Pécs, 7624 Pécs, Hungary; 4Institute of Physiotherapy and Sports Science, Faculty of Health Science, University of Pécs, 7621 Pécs, Hungary; 5Sport Science Research Group, Gál Ferenc University, 6720 Szeged, Hungary; 6Department of Physical Education Theory and Methodology, Hungarian University of Sports Science, 1123 Budapest, Hungary

**Keywords:** assessment tool, primary school, children, test battery, fundamental movement skill

## Abstract

This paper aimed to analyze fundamental movement skill (FMS) assessment tools that could be used for primary school children. In this narrative review, the Motoriktest für Vier- bis Sechjärige Kinder (MOT 4–6), Movement Assessment Battery for Children-2 (M-ABC-2), Motorische Basiskompetenzen (MOBAK) Körperkoordinationtest für Kinder (KTK), Test of Gross Motor Development (TGMD), Maastricht Motoriek Test (MMT) and the Bruininks–Oseretsky Test of Motor Proficiency-2 (BOT-2) were analyzed from a methodological perspective, such as the number of test items, tools and types of tests, in terms of the FMS area. The analysis revealed that to assess locomotor movement skills, the BOT-2 has an excellent test for running ability, but for detecting technical difficulties, the TGMD is recommended. To test hopping, the MMT is the best test. Object control movement skills are measured with throws, dribbles and catches. Most of the tools assessed these skills, but it turned out that the TGMD is the best for measuring object control. Stability movement skills are tested with static and dynamic balance tests. Dynamic balance is more frequently used, and the MOT 4–6, KTK and BOT-2 have the most tools to use. However, the MMT is an excellent test for static balance. Fine motor movement skills are easy to assess with the MMT and MOT 4–6, since they have low equipment requirements. The BOT-2 is the best measurement tool; however, it has high equipment requirements. All of the FMS assessment tools are good; however, we concluded that although these tools are excellent for research purposes, they are difficult to apply in a school setting. Thus, teachers and coaches are advised to always select a single task from the available assessment tools that is appropriate for the skills they would like to measure.

## 1. Introduction

Motor skills are fundamental abilities that enable individuals to perform various physical tasks efficiently [1]. Basic skills like walking, running and jumping and complex activities like sports and fine motor tasks play a crucial role in human development and daily functioning. Developing these skills in childhood is fundamental for competitive sports and lifelong activity, since this contributes to physical, mental and social development as well [2,3,4,5,6]. These benefits also highlight the importance of measuring and continuously monitoring these skills. An accurate motor skill assessment tool serves as an objective measure to evaluate an individual’s physical abilities, and it is essential for identifying strengths and weaknesses. Measuring motor skills in children is a priority for both physical education and youth sports, since it can help teachers and coaches to support the motor skills development that the individual requires.

Ideally, the pillars of motor skills should be developed before the onset of the rapid growth phase in adolescence, since previous studies have already demonstrated that childhood is the most sensitive period for the development of fundamental movement skills (FMSs) [5,7,8]. Several researchers refer to FMSs as the pillars of motor skills, since they are a set of foundational physical abilities that serve as building blocks for more complex and specialized movements. FMSs are essential for developing complex movements that involve the ability to move confidently and effectively in a wide range of physical activities. There are four main categories of fundamental movement skills: locomotor movement skills, object control movement skills, stability movement skills and fine motor movement skills [1,8].

Several researchers have stated the relevance of conducting FMS measurements on a regular basis [9,10]. The assessments also help identify strengths and weaknesses in coordination, balance, agility and other important skills [1]. Furthermore, a proper FMS assessment tool helps youth sports and rehabilitation; hence, it is used in various fields [11,12,13]. It is difficult to find a proper assessment tool for a specific skill. Several standardized FMS assessment tools were created in the past few decades to assess performance in early childhood [14,15,16,17]. Most of these tools aim at specific skills; thus, they contain different tasks, but all of them could help with early development, personalized training, performance optimization and injury prevention.

In our research, we investigated commonly used standardized FMS assessment tools based on previous studies that are appropriate for school-aged children [18,19,20]. We sought to identify those FMS assessment tools that are suitable for supporting motor learning in both physical education and youth sports by measuring motor indicators of young school children (4–10 years). We also aimed to help coaches and PE teachers to introduce appropriate motor development measuring tools, and help them choose the right tests to measure the abilities of their students or athletes. A study by Eddy and her colleagues [21] showed that most PE teachers and coaches are not aware of the FMS measures and development due to a lack of education; hence, we hope that this review can help them find the best assessment tools. Previous reviews have mainly focused on the psychometric properties of the assessment tools, such as their validity, reliability, types of measurement tools, etc. [19,22,23]. In our research, we examined assessment tools from a methodological perspective, such as their assessment time, test items, motor skills that the assessment tools measure, etc. Based on the previous studies, we analyzed the following assessment tools [18,19,20]: Motoriktest für Vier- bis Sechjärige Kinder (MOT 4–6), Movement Assessment Battery for Children-2 (M-ABC-2), Motorische Basiskompetenzen (MOBAK), Körperkoordinationtest für Kinder (KTK), Test of Gross Motor Development (TGMD), Maastrichtse Motoriek Test (MMT), Bruininks–Oseretsky Test of Motor Proficiency-2 (BOT-2); the TGMD and BOT-2 tests have a short form as well.

### Short Description of Movement Skill Assessment Tools

The MOT 4–6 assesses motor developmental status and can be used for the early detection of FMS delays or deficits [24,25]. The M-ABC-2, which is similar to the MOT 4–6, measures the developmental level of FMSs, and it can effectively screen for motor deficits and arrested motor development [15,26]. The MMT aims to objectively assess the qualitative aspects of motor skill patterns in addition to quantitative motor skill performance [27]. It is the only test that assesses both the execution and performance of the tasks. In the measurement of fine and gross motor skills, the authors claim that children at risk for attention deficit hyperactivity disorder (ADHD) can also be successfully filtered [27]. The KTK test primarily assesses the dynamic balance of the body. In addition to healthy individuals, it can be used for children with learning and behavioral difficulties, but it has been applied to children with brain damage as well [14,28]. The TGMD test measures the quality of movement patterns in the FMS. There are three different versions of this test [16,17,29,30]. It is an excellent test for assessing the impact of movement programming and developmental exercises [16,17]. The full and short forms of the BOT-2 can be used to assess fine and gross motor skill levels and their development. It can also be used for identifying individuals with mild to moderate motor coordination disorders. [31,32]. The MOBAK-1 and MOBAK-3 tests have been adapted to the curriculum, and assess life-stage specific basic skills that are necessary for children to be active and fit for sports. [33,34]. Finally, it must be acknowledged that the TGMD-3 and BOT-2 have a “short form”, which is an excellent tool for screening motor competence in young school children in a pragmatic and time-efficient manner [32,35].

## 2. Materials and Methods

We present a narrative overview on motor assessment tools [36]. We performed a non-systematic search in the PubMed, Scopus and Web of Science databases in May of 2023. We used the following terms: “Motor competence assessment” and “psychomotor performance test”. The only eligible criterion was to find studies that use any motor assessment tool. We did not select any specific date; the years varied between 1976 and 2023. Overall, 44 studies were found to meet this criterion, but we did not analyze these valuable articles in this study. During our analysis, we realized a systematic review is not an option since the basic descriptions of these assessment tools are published elsewhere, and they were not added into these databases. However, this search helped us to find the appropriate motor skill assessment tools. Finally, we searched the original guidelines of assessment tools based on these studies. Hence, the MOT 4–6, M-ABC-2, MOBAK, KTK, TGMD, TGMD-short form, MMT, BOT and the BOT-short form were analyzed. In the last step, we performed a methodological analysis of their characteristics (e.g., age group, assessment time, test items, etc.), and the types of subtests included in the assessment tools. Finally, we investigated the types of motor skills that these tools measure (e.g., gross motor skills). The contributions were collected by Á.V.N., T.B., M.D. and M.W.; then, they were critically reviewed by Á.V.N., F.GY. and T.B. All of the authors approved the final version.

## 3. Methodological Analysis of the Assessment Tools

Table 1 presents the characteristics of the assessment tools. High standard deviations are seen for both the quantity of test items (SD = 20.44) and the duration of the test (SD = 11.30). Three of the tools (MOT 4–6, MMT and BOT-2) appear to be results-oriented, while the M-ABC-2 and TGMD are more process-oriented. Only the TGMD has 2-level scaling in the evaluation process, i.e., evaluating correct or incorrect execution. The other process-oriented assessment tool uses a scale with three or more levels to evaluate partially accurate execution. During testing, the age-appropriate standardized scores from the result-focused assessment tools are used from raw performance scores. The instrument requirements of tools also differ widely, but all of them need some kind of sports equipment to produce the test. There are culture-specific differences. For example, the TGMD can also be performed with tennis and baseball [17]. Furthermore, testing the KTK requires specially manufactured equipment, such as a beam. To carry out the other tests (M-ABC-2, MMT, MOT 4–6, BOT-2), in addition to sports equipment, a table and other specific types of equipment are needed. We highlight the strengths and limitations for the assessment tools as well. One of the strengths of the M-ABC-2, BOT-2 and BOT-2 short form assessment tools are that they contain all of the FMS areas. Some of them can be used easily in education (MOT 4–6; MOBAK-1; MOBAK-3). The main issue with assessment tools varied from not including a certain area to taking too much time [19,22,37,38,39,40].

The content components of the FMS assessment tools are shown in Table 2. Locomotion movement analysis was measured with running tasks in three assessment tools (BOT-2; TGMD-2; TGMD-3). Additionally, most assessment tools involve skipping to measure locomotion movement skills. The tasks include one leg and/or two legs, which can be performed either in one spot or in a forward motion. Jumping tests are also included in half of the tools for locomotion movement analysis.

Measuring object control movement skills includes different types of throwing, dribbling, kicks, strikes, and other complex exercises (i.e., throw and catch). As seen in Table 2, most of the assessment tools measure a skill with one or more tasks; however, the KTK did not include any tests to measure object control movement skills. Furthermore, it is mostly dominated by upper-limb tests, but the MOBAK, TGMD and MMT examine lower leg coordination.

Static and dynamic balance can be found in the analyzed tools. Almost half of the tests include one-leg and/or two-leg static balance tasks executed with eyes open or closed. Except for the TGMD, all of the assessment tools include dynamic balance tasks, such as walking forward and backwards, and walking heel-to-toe on a walking line or balance beam. Only the MMT and BOT-2 assess fine motor skills without equipment (see Table 2). The test includes mainly tasks involving fingers and hands. The assessment tools assess these skills with equipment, such as tennis balls, folding paper, scoring cards, pens, etc.

Finally, we examined the assessment tools according to the number of types of tests included in the assessment tools as well (Table 3). The MMT had the highest number of tests (70 tests), but it did not measure locomotion movement skills. On the other hand, the BOT-2 had the most tests to measure that skill. The TGMD, MMT and BOT-2 had the most tests for measuring object control movement. The MMT also includes the highest number of balance tests (static = 14; dynamic = 20). The TGDM did not include any balance tests, and the MOBAK only measured dynamic balance. Assessing fine motor movement skills, the BOT-2 (24 tests) and MMT (28 tests) have the highest number of tests. Interestingly, the KTK test assesses dynamic balance only.

## 4. Discussion

In this study, we analyzed fundamental movement skill assessment tools from a methodological perspective. We examined the aspects that are necessary to assess their applicability in schools and sports clubs, including the number of test items, the measuring tools needed for the tests, and the time required for the assessments. The goal was to see which tests of the assessment tools can be used effectively to measure any FMS. Finally, we analyzed the content of the tools based on the FMSs that are measured. Overall, eleven assessment tools were analyzed in this study.

In investigating locomotor movement skills measurement, we found that the BOT-2 running test is excellent for testing running ability [32] since they use “shuttle run” tests for running agility. For detecting deficiencies in technical execution, the TGMD-2 or TGMD-3 are recommended to use [17]. Both tools help to evaluate running techniques with a unitary criteria system. Those coaches or PE teachers who want to investigate the effectiveness of running with the “moving variably” test of the MOBAK-3 will find it useful, in which the running and lateral running should be alternated for effective performance [41]. Almost all of the assessment tools include hop tests. Depending on the needs, one can choose between one-leg, two-legs, on-the-spot, and forward hopping. The MMT test is recommended [27] for testing the coordination of two body halves like the hands and feet. Assessing crossed movements during the hop, the BOT-2 assessment tool would be favorable, including suitable tasks [32]. The MOT 4–6 and KTK are recommended for the high jump, the MMT for the long jump, and the BOT-2 and TGMD for the long jump from a stationary position. The MOBAK is applicable for assessing forward rolls, and the MOT 4–6 test is important for considering the measurement of rolling around the longitudinal axis [27]. Since previous research has demonstrated a deficit in motor skills in school-aged children, some of the early childhood tasks in school-based assessments [42,43,44,45] should be considered for use, for example, the test of “rolling around the longitudinal axis” and tests of crawling and climbing as well.

The throws had a prominent place for object control movement skills. The tasks mainly assess the effectiveness of a target throw, such as a one-handed overarm, or a one-handed underarm throw. The TGMD has a good tool that assesses the quality of the execution of the throw. The two-handed catching skill appears in a variety of different tools. The children were tested with balls, hoops, and bean bags in the different assessment tools. Some tools offer complex tests such as “throw and catch” and “release and catch”; these can be found in the BOT-2 and MOBAK-3. The importance of object control movement skills has been shown in ball games and other sport techniques such as tennis and badminton [46,47,48]. In investigating object control movement skills, it turned out that boys are performing better in one-handed overarm throws, ball catching, and ball kicking. However, girls performed better in locomotion movement and positional changing skills (i.e., dynamic balance, jumping) [49,50,51,52,53].

Stability movement skills were tested in static and dynamic balance tests. Static balance is only assessed in four assessment tools. One-foot balancing, such as a flamingo test, can be performed on the floor, on a line, or on a beam, with eyes open and eyes closed. However, to compare the static stability of the two sides, we recommend the use of the MMT [27]. The KTK assesses dynamic balance; therefore, if we only want to explore this skill, the KTK is recommended. Dynamic balance is the most often used skill that other assessment tools measure. The tests for the MOT 4–6, KTK and BOT-2 are the best assessment tools to use. For example, we can find tests for walking forward and backwards, and for walking heel-to-toe on a walking line or balance beam, or through barriers. Furthermore, these balance tests can be performed on a narrower surface to challenge the students and athletes. Depending on our goal, the test could be appropriate for a sports team and for classes as well [14,27,32].

Fine motor movement skills are tested without any tools (e.g., “Hand tapping”, “Opposition of fingers and thumb”, “Pronation–supination”) in the MMT and BOT-2, and with a tennis ball in the MOT 4–6. Both physical education classes and training sessions can use these exercises. Other tasks involve assessing fine motor movement skills while seated at a table, indicate that these measurement parameters are not specifically connected to sports and physical activity [15,24,27,32].

Overall, the analysis revealed that to assess locomotor movement skills, the BOT-2 has an excellent test for running ability, but for detecting technical difficulties, the TGMD is recommended. To test hopping, the MMT has the best tests. Object control movement skills are measured with throws, dribbles and catches. Most of the tools assessed these skills, but it turned out that the TGMD has the most tests for it. Stability movement skills are tested with static and dynamic balance tests. When dynamic balance is more used, the MOT 4–6, KTK and BOT-2 have the most tools available. However, the MMT is an excellent test for static balance. Fine motor movement skills are easy to assess with the MMT and MOT 4–6, since they have low equipment requirements. The BOT-2 is the best tool for measuring; however, it has high equipment requirements.

The assessment tools created from different theoretical backgrounds often use different test items to assess similar skills [22]. Furthermore, we believe it is not necessary to choose a complex test battery for school classes or training, since it is advisable to select tests that are appropriate for the current curriculum or learning objective that are usable for both exploratory and summative assessments. However, the assessment tools in their full forms may be better for research purposes. Furthermore, FMS assessment can be used to diagnose the movement development level of your own team/athletes, and then adjust the necessary training programs accordingly to the results. We believe that PE teachers and coaches must be aware that a particular motor skill varies widely according to both age and the individual [54,55]. There is no predetermined strict order in which these skills should be developed, despite the fact that some skills are undoubtedly easier to acquire, and the sequence of major movements is determined by the course of human development [55]. However, lack of proper FMS assessment could cause life-long difficulties in learning motor skills [8]. If we do not correct or make up for the deficiencies in time, we can significantly impair the effectiveness of motor learning and movement control in all areas of physical abilities.

Research by Pienaar [54] revealed that specific FMS development varies across nations and social contexts. Thus, there is no specific test battery that PE teachers and coaches can use and implement; however, it is important, since recent research has shown that basic skills are missing, even in youth sports, which could lead to injuries as well [56].

We believe this review will help professionals to understand more about FMS assessment tools; therefore, future research goals should focus on reviews of FMS concepts, and initiate a drive towards uniformity to create a motor skills test system based on educational curricula, following the example of the MOBAK assessment tool.

## 5. Conclusions

Assessment tools designed to assess FMSs in school-aged children are excellent for research purposes, but they are difficult to apply in a school setting and to sports teams. Thus, teachers and coaches are advised to always select a single task from the available assessment tools that is appropriate to the curriculum and supports the teaching and learning process. As there are large social and cultural differences in the definition of both FMS content and age-related performance, the focus should be on individual tests that help to find the motor skill gap that a child may have. Furthermore, learning about individual differences in test scores allows differentiation, and supports individual learning pathways in the acquisition of FMSs, which is an essential guarantee of catching up, developing motor skills, and thus preparing for a healthy life and nurturing sports talents.

In conclusion, we recommend this review for academics who are teaching future coaches and PE teachers, and promote FMS assessment with help in choosing test items that are suited to their educational goals.

## Figures and Tables

**Table 1 sports-11-00178-t001:** Characteristics and theoretical frameworks of movement skill assessment tools.

Name	Aim	Age	Time	Items	Result Oriented	Process Oriented	Evaluation	Devices	Strengths	Limitations	Citations
Motor skills test for 4–6-year-old children (MOT 4–6)	Early detection of FMS delay or deficiency	4–6 years	20–25 min	18 items	Yes	Yes	0–2 points/Items–raw score	Also requires sports equipment and special equipment	Can be used in an educational environment. A quality assessment is also possible. The measurement can also be done in the classroom.	It does not include a static balance task. It requires several special tools. It contains several similar tasks, thereby increasing the measurement time.	(Zimmer and Volkamer, 1987 Zimmer, 2006;) [24,25]
Movement Assessment Battery for Children (M-ABC-2)	Detection of delay or deficiency	3–16 years/3 age bands	20–30 min	8 items	No	Yes	0–5 point/Items	Also requires sports equipment and special equipment	All test areas are included. Cross-cultural validity. Few tasks, little assessment time.	It requires several special tools.	(Henderson and Sugden, 1992; Henderson, Sugden and Barnett 2019) [15,26]
Maastrichtse Motoriek Test (MMT)	To evaluate the quantitative and qualitative components of movement at the same time.	5–6 years, kindergarten school transition	30 min	70 items	Yes	Yes	0–2 points/Items	Requires sports equipment	It also includes result- and process-oriented assessments evaluation. It places great emphasis on the evaluation of speed coordination. It also measures sense of rhythm.	There are too many tasks. Time consuming.	(Vles et al., 2004) [27]
Bruininks–Oseretsky Test of Motor Proficiency (BOT-2)	Fine and gross motor skill levels and suitable for identifying movement coordination disorders.	4–21 years	45–60 min	53 items	Yes	No	Ranging from a 2-point scale to a 13-point scale	Also requires sports equipment and special equipment	You can choose composites or necessary subtests. All test areas are included. The measurement of fine motor skills is emphasized.	There are too many tasks that are tiring for young children. Time consuming. It requires several special tools.	(Bruininks, 1978; Bruininks and Bruininks, 2012) [31,32]
Bruininks–Oseretsky Test of Motor Proficiency–Short form	Screening test	4–21 years	15–20 min	14 items	Yes	No	Ranging from a 2-point scale to a 13-point scale	Also requires sports equipment and special equipment	All test areas are included.	It requires several special tools.	(Bruininks, 1978; Bruininks and Bruininks, 2005) [31,32]
Körperkoordinationtest für Kinder (KTK)	Screening dynamic balance skills with typical or brain damage, behavioral problems or learning difficulties children.	4–14 years	20 min	4 items	Yes	No	Raw scores/standardized scores	Requires special sports equipment	It differentiates well from light to heavy.	The test only measures the ability of dynamic balance. It requires several special tools.	(Kiphard and Shilling, 1974; Kiphard and Schilling, 2007) [14,28]
Test of Gross Motor Development-2 (TGMD-2)	Backlog in gross motor performance	3–10 years	15–20 min	12 items	No	Yes	0–1 point/item	Requires sports equipment	Excellent for evaluating movement quality.	No stability subtest. Culturally dependent.	(Ulrich, 1985; Ulrich, 2000) [16,17]
Test of Gross Motor Development-3 (TGMD-3)	Backlog in gross motor performance	3–10 years	17–22 min	13 items	No	Yes	0–1 point/item	Requires sports equipment	Excellent for evaluating movement quality.	No stability subtest. Culturally dependent.	(Webster and Ulrich, 2017) [29]
Test of Gross Motor Development-3 Short form (TGMD-3 Shord form)	Backlog in gross motor performance	3–10 years	10–13 min	7 items	No	Yes	0–1 point/item	Requires sports equipment	Excellent for evaluating movement quality.	No stability subtest. Culturally dependent.	(Duncan et al., 2022) [30]
Motorische Basiskompetenzen (MOBAK-1)	Screen the level of student’s motor competence	6–7 years	10–12 min	8 items	Yes	No	0–2 point/item	Requires sports equipment	The subtests are age-specifically adapted to the curriculum requirements of physical education. It can be used well in PE lessons. Uses appropriate equipment in PE.	Time-consuming: 5 children can be assessed during a 45-min PE lesson.	(Herrmann et al., 2019) [33]
Motorische Basiskompetenzen (MOBAK-3)	Screen the level of student’s motor competence	8–9 years	10–12 min	8 items	Yes	No	0–2 point/item	Requires sports equipment	The subtests are age-specifically adapted to the curriculum requirements of physical education. It can be used well in PE. lessons. Uses appropriate equipment in PE.	Time-consuming: 5 children can be assessed during a 45-min PE lesson.	(Herrmann and Seelig, 2017) [34]

**Table 2 sports-11-00178-t002:** Content analysis of FMS assessment tools.

Subtests/Tasks	MOT 4–6	M-ABC-2 (3–6 Age)	M-ABC-2 (7–10 Age)	MMT	BOT-2	BOT-2 Short Form	KTK	TGMD-2	TGMD-3	TGMD-3 Short Form	MOBAK-1 (6–7 Age)	MOBAK-3 (8–9 Age)
*Locomotion Motor Movement Skills*												
Run					Shuttle run			X	X			
Hop with 1 leg	X		Forward	Left, Right, Forward left leg, Forward right leg	X	X		X	X	X		
Hop with 2 legs	Jumping Jack	Forward		X	Jumping Jack, Same side synchronized, Opposite side synchronized	Same side synchronized						
Hop with 1 and 2 legs											Forward	
Gallop								X	X	X		
Slide								X	X		X	
Run and slide												X
Leap/Skip								X	X			
High jump	X						X					
Long jump				X								
Long jump from place					X			X	X	X		
Side hop					One-legged, Two-legged							
Rolling around longitudinal axis	X											
Rolling forward											X	X
Knee push ups					X	X						
Sit up					X	X						
Wall sit					X							
V-up					X							
Hiding through hoops	X											
Complex exercise: Stand up–sit down	X											
*Object Control Movement Skills*												
Throw overhand	Target				Target			X	X	X	Target	Target
Throw underhand		Target	Target						X			
Catch (two-handed)	Stick, Ring	Beanbag	X	X				X	X	X	X	
Dribble stationary				X	One hand, Alternate hand	Alternate hand		X	X	X		
Dribble forward											X	Slalom
Dribble with leg											X	Slalom
Kick				Right leg, Left leg				X	X			
Strike								X	One hand, Two hands	Two hands		
Underhand roll								X				
Rope skipping												X
Complex exercise: Throw and catch					One hand, Two hand							Two hands
Complex exercise: Drop and catch					One hand, Two hand	Both hands						
*Stability Movement Skills*												
Static balance skills												
One-leg balance		X	X	Right leg, Left leg	Eyes open, Eyes closed	X						
Stork stand				Right leg, Left leg								
One-leg balance on a beam					Eyes open, Eyes closed, Heel-to-toe							
Two-leg balance on the line					Eyes open, Eyes closed							
Two-leg balance				Eyes closed, Eyes closed arms forward								
Standing on toes				X								
Dynamic balance skills												
Walk on heels				X								
Walk on the toes				X								
Walking forward on the line	X	Heels raised	Heel-to-toe	Tightrope walker	X, Heel-to-toe	X					Beam	Beam, Barrier
Walking backwards on the line	X						Beam					
Jumping sideways over a slat	Rope				Beam		X					
Moving sideways							X					
Twisting jump in the hoop	X											
Complex exercise: Jump and one-leg balance	X											
*Fine Motor Movement skills*												
*Without equipment*												
Hand tapping				Left, Right								
Feet tapping				Left, Right								
Tapping feet and fingers					Same side synchronized; Opposite side synchronized	Same side synchronized						
Pivot thumbs and index fingers					X							
Touching nose with index fingers—eyes closed					X							
Pronation–supination				Dominant hand, Non-dominant hand, Both hands								
Opposition of fingers and thumb				Dominant hand, Non-dominant hand, Both hands								
*With equipment*												
Put tennis balls in boxes	X											
Drawing points	X				X							
Drawing lines		X	X		2 items	X						
Copy					8 items	2 items						
Match packing	X											
Grip with toes	X											
Posting coins		X			X	X						
Threading beads		X										
Folding paper					X	X						
Placing pegs			X									
Threading lace			X									
Placing pegs in pegboard					X							
Soring cards					X							
Stringing blocks					X							
Filling in a circle					X							
Filing in a star					X							
Connecting dots					X							
Cutting out a circle					X							
Pen				Dominant hand, Non-dominant hand								

**Table 3 sports-11-00178-t003:** Subareas of movement skill assessment tools.

*Assessment Tools*	*Locomotion Movement Skills*	*Object Control Movement Skills*	*Stability Movement Skills*		*Fine Motor Movement Skills*	All Tests
			Static Balance Skills	Dynamic Balance Skills		
**MOT 4–6**	6	3	.	5	4	18
**M-ABC-2 (3–6 age)**	1	2	1	1	3	8
**M-ABC-2 (7–10 age)**	1	2	1	1	3	8
**KTK**	.	.	.	4	.	4
**TGMD-2**	6	6	.	.	.	12
**TGMD-3**	6	7	.	.	.	13
**TGMD-3 Short form**	3	4	.	.	.	7
**MMT**	.	8	14	20	28	70
**BOT-2**	12	7	7	3	24	53
**BOT-2 Short form**	4	2	1	1	6	14
**MOBAK1.**	3	4	.	1	.	8
**MOBAK3.**	3	4	.	1	.	8

## Data Availability

There were no data used in this study.

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
