# Peer review of "Assessment Tools Measuring Fundamental Movement Skills of Primary School Children: A Narrative Review in Methodological Perspective"

_sports, 2023, doi:10.3390/sports11090178_

Round 1

Reviewer 1 Report

"Assessment tools measuring motor skills of primary school children: A narrative review in methodological perspective" by Nagy and colleagues is an interesting narrative review with the aim of identifying which assessment tools for motor skills could be used for primary school children. 

The topic is very interesting and of current interest to the scientific community. The manuscript is well organized and well-structured, and the bibliographical references are also appropriate. I only suggest that the authors improve the abstract, as this section is of paramount importance as it should entice the reader to read the entire work. In this case, although the authors have described the content of their manuscript in detail, in my opinion it is incomplete. Indeed, it would be appropriate to add a background in a concise manner that well introduces the subject matter of the manuscript. Furthermore, the abstract lacks a concluding sentence suggesting possible future perspectives that might emerge from the results obtained.

Furthermore, I suggest the authors revise the English language. 

Moderate editing of English language required.

Author Response

First, we would like to thank to the reviewer responses to helped us improve this manuscript. All suggestions were considered. Please note the manuscript were significantly change, hence the introduction and the discussion were expanded, and the abstract was revised. Furthermore, we changed the title to highlight FMS and we did it as well in the whole manuscript. All change were marked as red in the manuscript.

            We would like to thank you your comment regarding the abstract. We agree with you, therefore we expanded and add information about our result in it. Furthermore, the English of the manuscript were revised.

Reviewer 2 Report

The paper concerns a current and very complex topic due to its effects on teaching methodologies. The work is well structured and organic; references are up to date. The methodological indication of using only some motor tasks in the school curriculum is appropriate. 

However, the future developments of the study do not emerge; eg: a) the indication of the protocols used in adapted physical activity; b) relationships between the assessment of motor skills and academic performance.

Author Response

First, we would like to thank to the reviewer responses to helped us improve this manuscript. All suggestions were considered. Please note the manuscript were significantly change, hence the introduction and the discussion were expanded, and the abstract was revised. Furthermore, we changed the title to highlight FMS and we did it as well in the whole manuscript. All change were marked as red in the manuscript.

Thank you for your kind word and your suggestions regarding our manuscript. We considered your suggestion, but adapted physical activity were not involved in this study, since these assessment tools are eligible for normal and adapted PE as well and we do not want to highlight one or other. The relationships between the assessment of motor skills and academic performance were not involved either, since this narrative review only concentrated the methodology of the assessment tools

Reviewer 3 Report

Thank you to the authors and journal editor for allowing me to review this paper. Overall, this paper proposes an good rationale and presents some useful summary information to the reader. However, it does not present novel findings, and the discussion and introduction could be much improved. I would recommend the authors spend further time ensuring the aim of the research paper is presented clearly and appropriately reached in the discussion.  

Abstract

Line 24: Even though several assessment tools were analyzed concluded that these tools are excellent for research purposes but are difficult to apply in a school setting

Change to: Even though several assessment tools were analyzed, WE concluded that these tools are excellent for research purposes but are difficult to apply in a school setting

Line 27: We hope with the help of this review 27 PE teachers and coaches could easily choose the right test for a specific motor skill

Although it would be great if this was the reality for PE teachers and coaches, further dissemination and work from these results would need to take place for this to occur, as it is unlikely that teachers will read academic papers such as this. Perhaps some more detail on how this could be achieved would be suitable here.

Introduction

Overall, this introduction is well written and provides a decent rationale for the review taking place. Further detail about areas such as the ‘pillars of motor skills’ would enhance the readers understanding. More information about what schools and sports clubs need from motor skill assessments would also be of use- what would make these tests easy for them to use and implement within their setting? Further comments are made below:

 Line 42: take out ‘as well’, it is not needed here for the sentence to make sense.

Line 48: what are these ‘prevention-orientated measures?’

Line 48-53:  this section of the introduction feels confused, as if many separate statements about motor skill development and measurement are being made, but these do not fit or flow together. This needs to be rewritten to ensure there is a clear rationale for the study and why motor development measurement needs to take place. Although there may be various needs for measurement, it may be better to focus on one or two specific areas such as removing information about athletic training, when focussing on early childhood.

Line 61: FMS is suddenly introduced to the reader in this line. FMS should be given introduction earlier in this section, so the reader is aware of how these are related to motor skills.

Line 68: when introducing the methodological perspective, it may be worth mentioning what this might involve, what elements of the tools etc. So a reader can understand the difference between psychometric properties.

Section 1.1: I would like to commend the authors on this section, it is well written and clearly describes which each tool is, and how they differ from one another and their intended uses. Congratulations.

Materials and Methods

More information about why a systematic review was not suitable would have been appreciated within this section. There’s feels to be a lack of rationale about why the chosen methods were chosen. Has there been a review that has performed this in a similar way before- e.g. the methodological perspective of analysis.

Methodological analysis

Line 114: what is time measured in? please give the units.

Table 1. This is a great summary table, which gather lots of useful information. I think this is something that could be disseminated further to help coaches and teacher mentioned in the abstract.

In terms of identifying strengths and limitations mentioned in these tables, I feel it would be appropriate to support some of these with citations of literature that report the same findings e.g. taking too long, or being too tiring for children, otherwise this completely personal opinion.

Within the Age column it should 4-6 YEARS rather than AGE.

Line 146: Remove ‘Most of’

Table 2: One each new page of the table the row with all the different tools/tests needs to be included at the top to aid the reader in understanding and using the table, otherwise it is very hard to follow.

Line 152:  We EXAMINED

Discussion

This is discussion could be enhanced to improve the quality of this paper. There is little discussion of why the skills are important to children and how they may enhance their overall movement ability and future health and development. There is also little discussion of how and why a teacher may compile different skills from different tools into a lesson or session, there should be some thought around this.

Line 170-172: Mention why these tools are the best for measuring this, don’t just assume the reader will know why.

Line 177-180: please review this sentence for clarity, I’m not sure I understand what is meant.

Line 195-198: Although these findings are important they don’t highlight why object control skills are important, they simply show that there are sex differences in the performance of object control and locomotor skills. The sentence following line 194 need to be written again.

Line 210: examples of fine motor skill measurement without tools would be useful to the reader here, so they know what they might include.

Line 228: what is the National Core Curriculum?

Conclusions

Line 235:  FMS has not clearly been defined to the reader, so this sentence may be hard to interpret.

Line 236: this is the first-time intervention programmes have been mention, I feel the focus should stay on the assessment tools.

Line 239: this is the first time ‘movement literacy’ has been used in the paper. I would recommend removing this, as it currently has no context within this work.

Again, in this section, find consistency with motor skills or FMS rather than swapping between the two, as this is very confusing to read.

Minor editing of English is needed and detailed in comments above. 

Author Response

First, we would like to thank to the reviewer responses to helped us improve this manuscript. All suggestions were considered. Please note the manuscript were significantly change, hence the introduction and the discussion were expanded, and the abstract was revised. Furthermore, we changed the title to highlight FMS and we did it as well in the whole manuscript. All change were marked as red in the manuscript.

            Please see our detailed answer below:

Abstract

Line 24: Even though several assessment tools were analyzed concluded that these tools are excellent for research purposes but are difficult to apply in a school setting

Change to: Even though several assessment tools were analyzed, WE concluded that these tools are excellent for research purposes but are difficult to apply in a school setting

Thank you, we changed it.

Line 27: We hope with the help of this review PE teachers and coaches could easily choose the right test for a specific motor skill

Although it would be great if this was the reality for PE teachers and coaches, further dissemination and work from these results would need to take place for this to occur, as it is unlikely that teachers will read academic papers such as this. Perhaps some more detail on how this could be achieved would be suitable here.

Thank you for highlighting this issue. We are agreeing with you in this manner. Therefore, we added a final thought about this in the conclusion and we revise it in the abstract.

Introduction

Further detail about areas such as the ‘pillars of motor skills’ would enhance the readers understanding.

Thank you for your comment regarding this issue. We concluded that we highlight the importance of the pillars of motor skills. Hence, we added more information about Fundamental Movement skills to the introduction and we highlighted in the title as well.

More information about what schools and sports clubs need from motor skill assessments would also be of use- what would make these tests easy for them to use and implement within their setting?

Thank you for your recommendation. We added the following: Line: 73-75

A study by Eddy and his colleagues (21) showed that most of the PE teachers and coaches does not aware of FMS measures and development, hence we hope this review can help them to find the best assessment tools.

Line 42: take out ‘as well’, it is not needed here for the sentence to make sense.

We deleted 'as well'

Line 48: what are these ‘prevention-orientated measures?’

Thank you for our comment. We deleted this sentence to the clearer understand.

Line 48-53:  this section of the introduction feels confused, as if many separate statements about motor skill development and measurement are being made, but these do not fit or flow together. This needs to be rewritten to ensure there is a clear rationale for the study and why motor development measurement needs to take place. Although there may be various needs for measurement, it may be better to focus on one or two specific areas such as removing information about athletic training, when focusing on early childhood.

Thank you for your comment, we reconstructed as following: It also helps identify strengths and weaknesses in motor coordination, balance, agility, and other important skills [1]. Furthermore, a proper motor skill assessment tool helps youth sports and rehabilitation; hence it is used for various fields [11–13].

Line 61: FMS is suddenly introduced to the reader in this line. FMS should be given introduction earlier in this section, so the reader is aware of how these are related to motor skills.

Thank you for your comment we added FMS to line 46-51

Several researchers referring FMS as the pillars of motor skills, since it is set of founda-tional physical abilities that serve as building blocks for more complex and specialized movements. FMS are essential for developing complex movements, which is the ability to move confidently and effectively in a wide range of physical activities. There are four main categories of fundamental movement skills: Locomotor movements skills, object control movement skills, stability movement skills, fine motor movement skills

Line 68: when introducing the methodological perspective, it may be worth mentioning what this might involve, what elements of the tools etc. So a reader can understand the difference between psychometric properties. 

Thank you for your comment: We added the following: such as the assessment time, test item, motor skills that the assessment tools measures etc. (line 66-67)

Materials and Methods

More information about why a systematic review was not suitable would have been appreciated within this section.

During our analysis we realized systematic review is not an option since the basic de-scription of these assessment tools are published elsewhere and they were not adding into these databases. However, this search helped us the find the appropriate motor skill assessment tools

There’s feels to be a lack of rationale about why the chosen methods were chosen. Has there been a review that has performed this in a similar way before- e.g. the methodological perspective of analysis.

Thank you for your comment we added more information about methodology.

Methodological analysis

Line 114: what is time measured in? please give the units.

Table 1. This is a great summary table, which gather lots of useful information. I think this is something that could be disseminated further to help coaches and teacher mentioned in the abstract.

Thank you for your advice. It is hard to put every detail to the abstract, however we agree that it is an important table to demonstrate the assessment tools. Therefore, we highlighted in the aspects in the abstract and we reflect more on this in the discussion.

In terms of identifying strengths and limitations mentioned in these tables, I feel it would be appropriate to support some of these with citations of literature that report the same findings e.g. taking too long, or being too tiring for children, otherwise this completely personal opinion.

Thank you for your comment! We added the citations, and we added some text as well to make strength and limitations on overlook besides the table (line: 139-144)

We highlight the strengths and limitations for the assessment tools as well. One of the strengths of M-ABC-2, BOT-2 And BOT-2 short form are contain all the FMS areas. Some can be used easily in education (MOT 4-6; MOBAK-1; MOBAK-3). The main issue with assessment tools is varied between does not include a certain area or take too much time  [19,22,37–40].

Within the Age column it should 4-6 YEARS rather than AGE.

Thank you for your comment. we changed age to years

Line 146: Remove ‘Most of’

It was removed

Table 2: One each new page of the table the row with all the different tools/tests needs to be included at the top to aid the reader in understanding and using the table, otherwise it is very hard to follow.

Thank you for your comment. We add the first raw to each page.

Discussion

There is little discussion of why the skills are important to children and how they may enhance their overall movement ability and future health and development.

Thank you for your comment to this issue. We added the following: line 235-238

However, developing FMS could cause life-long difficulties on learning motor skills [1]. If we do not correct or make up for the deficiencies in time, we can significantly impair the effectiveness of motor learning and movement control in all areas of physical abilities.

There is also little discussion of how and why a teacher may compile different skills from different tools into a lesson or session, there should be some thought around this. 

Thank you for commenting we revised the discussion enterally to make our point. I recommend to  line 250 - 259 to this comment.

Line 170-172: Mention why these tools are the best for measuring this, don’t just assume the reader will know why.

Thank you for your comment on this issue. We added more information on this. Please see line 187-190.

Investigating measuring Locomotor Movement Skills, we found that the BOT-2 running test is excellent for testing running ability [31], since they use shuttle run tests which is use for running agility. Detecting deficiencies in technical execution, TGMD-2 or TGMD-3 are recommended to use [17]. Both tools help to evaluate running techniques in a unitary criteria system.

Line 177-180: please review this sentence for clarity, I’m not sure I understand what is meant.

Thank you for your comment we review and changed this sentence for clarity

Line 195-198: Although these findings are important, they don’t highlight why object control skills are important, they simply show that there are sex differences in the performance of object control and locomotor skills. The sentence following line 194 need to be written again.

Thank you for your comment. We revied the sentence, hoping it's clear for now.

The importance of Object Control Movement Skills has been showed in ball games and other sport techniques such as tennis and badminton. [46–48]. Investigating Object Control Movement Skills it turned out that the boys are performing better in one-handed overarm throw, ball catching, and ball kicking. However, the girls performed better in locomotion movement and positional changing skills (i.e., dynamic balance, jumping) [49–53].

Line 210: examples of fine motor skill measurement without tools would be useful to the reader here, so they know what they might include.

Thank you for your comment we added example to the paragraphs.

Line 228: what is the National Core Curriculum?

 We change to educational curriculum, to make it clear for the readers

Conclusions

Line 235:  FMS has not clearly been defined to the reader, so this sentence may be hard to interpret.

We added this part in the introduction. Please see earlier.

Line 236: this is the first-time intervention programmes have been mention, I feel the focus should stay on the assessment tools.

Thank you for your note. We change intervention programs to "individual tasks" to increase the clarity.

Line 239: this is the first time ‘movement literacy’ has been used in the paper. I would recommend removing this, as it currently has no context within this work.

We change to motor skills.

Again, in this section, find consistency with motor skills or FMS rather than swapping between the two, as this is very confusing to read.

Thank you for your comment we used FMS consistently in the discussion.

Round 2

Reviewer 3 Report

Thank to the authors for throughly revising their manuscript in a short period of time and the responses to previous review comments. There have been several changes that enhance the article and make an clearer read, congratulations. The additional sections on FMS are particularly useful. 

There have been some substantial changes to the manuscript which have not been highlighted, please ensure this is done during future submissions to aid the reviewer in understanding all changes. Also ensure the lines provided on your response are correct, as this makes it easier to find the changes.

I have made some recommendations to improve the clarity of the new lines 23-31, changes are highlighted in red.

The analysis revealed that to assess Locomotor Movement Skills BOT-2 has an excellent test for running ability, but for detecting technical difficulties TGMD is recommended. To test hopping MMT has the best tests. Object Control Movement Skills are measured with throws, dribbles, and catches. Most of the tools assessed these skills, but it turned out TGMD has the most tests for measuring object control. Stability Movement Skills were tested with static and dynamic balance tests. Dynamic balance is more frequently used and MOT 4-6, KTK and BOT-2 has the most tools to use. However, MMT is an excellent test for static balance. Fine Motor Movements Skills are easy to assess with MMT and MOT 4-6, since it has low equipment requirements, but BOT-2 are the best to 30 tool for measure, however it has high equipment requirements.

Change Line 52: Several researchers REFER TO FMS as...

Change Line 74: A study by Eddy and his colleagues [21] showed that most of the PE teachers and coaches are not aware of FMS measures and development, due lack of education, hence we hope this review can help them to find the best assessment tools.
(Lucy Eddy is female for future reference)

Change Line 143: One of the strengths of M-ABC-2, BOT-2 And BOT-2 short form are that they contain all the FMS areas.

The main issue with assessment tools is varied between does not including a certain area or taking too much time [19,22,37–40].

There needs to be a thorough proof read of this work to ensure the english is sound and correct. 

Author Response

Dear Reviewer!

Thank you for the deep analysis of our manuscript. We thank you for all the effort in both review rounds. We made all the changes you ask in your review. Please note the changes were tracked changed in Microsoft Word.

Sincerely,

Authors.